# The contribution of vulnerability to emotional contagion to the expression of psychological distress in older adults

Marie-Josée Richer[1]*, Sébastien Grenier[2,3], Pierrich Plusquellec[1,4]

**1** Department of Psychoeducation, University of Montreal, Montreal, Canada, **2** Institut Universitaire de Gériatrie de Montréal Research Center, Montreal, Canada, **3** Department of Psychology, University of Montreal, Montreal, Canada, **4** University Institute of Montreal in Mental Health Research Centre, Research Center on Non-Verbal Communication Sciences, Montreal, Canada

* marie-josee.richer@umontreal.ca

## Abstract

This study examines the differential weight of a wide range of factors—sociodemographic factors, indicators of autonomy, social support, coping styles, vulnerability to emotional contagion, and empathy—in the presence of two profiles of psychological distress and in their absence. This cross-sectional study included 170 older adults. As assessed by the Hospital Anxiety and Depression Scale (HADS), 65.9% of the individuals in the sample had a clinical or subthreshold level of anxiety and depression (score > 1). Based on the HADS's clinical cutoff scores for the anxiety and depression subscales, three profiles were created for the no distress, anxiety, and anxious depression groups. The profiles did not differ on demographic indicators except for sex. Vulnerability to emotional contagion, satisfaction with the social network and coping styles emerged as factors weighing the likelihood of being in either of the psychological distress groups relative to individuals with no distress. After controlling for adversity and psychotropic treatment, vulnerability to emotional contagion had the strongest relationship with both psychological distress profiles. Future research, such as a prospective longitudinal study, may provide an opportunity to explain the direction of the relationship between psychological distress and the factors studied, particularly vulnerability to emotional contagion.

## Introduction

Psychological resilience has been described as the ability to adapt and maintain one's psychological health in the face of adversity throughout life [1]. Scholars have recently recognized the complexity of psychological resilience and, consequently, the multifactorial nature of the processes involved [2]. In a process-based framework, psychological resilience involves the individual's personal competencies and characteristics, the context, and the time in which they are displayed [1].

Most importantly, resilience occurs only in the presence of adversity. Lazarus and Folkman [3] described adversity as the presence of stressful events. Although stress is a normal part of

**Data Availability Statement:** The dataset generated and analyzed for the current study are

available from the Open Science Framework (OSF; DOI 10.17605/OSF.IO/3A95F).

**Funding:** This study was supported by a grant from Le Groupe Maurice (www.legroupemaurice. com). Sébastien Grenier is supported by a Fonds de Recherche du Québec -Santé (FRQS) Senior salary award. Marie-Josée Richer is supported by a Fonds de Recherche du Québec -Culture (FRQC) doctoral award. The funders had no role in study design, data collection and analysis, decision to publish, or preparation of the manuscript.

**Competing interests:** The authors have declared that no competing interests exist.

life, both the stress of daily life and major stressful events (e.g., acute events such as the loss of a loved one or a chronic event such as caregiving) are important risk factors for psychopathology across the lifespan [4]. In the face of adversity, psychological resilience equates to maintaining one's mental health or to not succumbing to a state of distress [5]. Psychological distress, contingent on a lack of psychological resilience, is defined as an emotional disturbance that affects the functioning of the person experiencing it [6]. This condition is largely characterized by mild to severe symptoms of depression and anxiety such as sadness, loss of interest, and feelings of tension [7]. We recognize that definitions of psychological distress vary widely across studies and may include a variety of symptoms. In this study, we refer specifically to the expression of anxiety and depression when discussing psychological distress. In later life, resilience is related to the prevalence and consequences of psychological distress, as well as to changes in the physiological response to stress when exposed to adversity.

## Prevalence and consequences of psychological distress in later life

Anxiety and depressive disorders are highly prevalent among older adults. They often occur as comorbid disorders and share many risk factors [8–10]. Attention is given not only to the clinical disorders themselves, but also to the subthreshold symptoms of anxiety and depression, as they are common and have similar consequences for this population [10,11]. Due to the characteristics of the selected older adult populations and the chosen breakpoint for assessing symptom severity, prevalence rates can vary significantly between studies, ranging from 6% to 52% [12].

Studies have shown that older adults who experience symptoms of anxiety and depression are at higher risk of developing health problems such as cardiovascular disease [13,14], diabetes [15], cognitive impairment [16–18], and even premature mortality [13,15,18,19]. They are also more likely to have mental health problems and functional limitations, as well as to report lower quality of life [20,21]. In a sequence of reciprocal cause and effects, some consequences —like perceived functional limitations—can also contribute to the adversity experienced and they may increase the severity of psychological distress [22].

## Coping with adversity in later life

In recent reviews, Goncharova [23] and Rao and Androulakis [24] reported age-related changes in the hypothalamic–pituitary–adrenal (HPA) axis, that regulates the stress response, suggesting a limited psychophysiological coping capacity in later life when faced with adversity. According to Epel et al. [25], adversity typically presents itself in the form of explicit and implicit stressors.

**Explicit adversity.** Explicit stressors include situations that are perceived as threatening. In contrast to absolute threats (i.e., those arising from life-threatening events), relative threats arise from a sense of lack of control over a situation, its unpredictability, its novelty, or a sense of threat to one's ego [26,27]. These relative stressors can be acute (e.g., being late for an appointment) or more insidious, such as daily annoyances (e.g., frequent arguments with a partner) or chronic stress (e.g., caregiving) [25]. Many relative stressful situations can occur, such as normal aging processes that affect daily functioning, health problems, the loss of a partner, moving into a retirement home, difficulties with children, isolation, or caregiving [28].

**Implicit adversity: The role of emotional contagion.** Individuals may also experience implicit stressors in interpersonal situations that trigger an automatic physiological stress response. Stress contagion is rooted in the phenomenon of emotional contagion [29], which can be defined as "the tendency to automatically mimic and synchronize facial expressions, vocalizations, postures, and movements with those of another person and, consequently, to

converge emotionally" [30 p153]. Physiological resonance is one of the main mechanisms involved. Buchanan et al. [31] demonstrated the contagion of the physiological response to acute stress between a target and an observer behind one-way glass. In their study, observers showed an increase in their cortisol response (i.e. of the stress hormone) that was proportional to that of the individuals undergoing social stress, illustrating the resonance of the stress response at the physiological level. Engert et al. [32,33] found a similar and even stronger resonance when the observer and the target were significant others. Emotional contagion is an emerging issue in stress resilience because it affects the ability to regulate emotions in the face of social stress [34]. From the perspective of personality differences, some individuals may be more prone to catch the emotions of others. This vulnerability to emotional contagion has been conceptualized as a trait-like disposition [35].

Vulnerability to emotional contagion is known to be influenced by a variety of factors, the most studied of which are relationship closeness, as noted above, and higher levels of empathy [33]. When emotional contagion occurs, individuals do not need to know the source of the shared state and actually experience it as their own. In contrast, empathy is a process of consciously inferring the affective state of another person, which allows for the recognition that the other person is the source of that affective state, not oneself [36]. Empathy is closely related to emotional contagion, but it can be divided into cognitive versus affective domains. The cognitive domain reflects the tendency to spontaneously adopt the psychological point of view of others [37]. The affective aspects of empathy, these include feelings of sympathy and concern for unfortunate others, and feelings of discomfort when confronted with intense interpersonal situations [37]. Compared to cognitive empathy, emotional empathy might facilitate emotional contagion and result in a more vigorous mirroring of others' states [38].

The concept of vulnerability to emotional contagion can be situated within a biopsychosocial theorical framework of stress. This understanding can enhance our comprehension of the implicit stress level that affects the body in a manner analogous to explicit stressors. To prevent the aforementioned outcomes and enhance psychological resilience in the context of ongoing daily life stress in later life, it is essential to recognize and understand the associated risk factors that can be translated into practice [10,39,40].

## Individual risk factors of psychological distress

To date, several studies have been conducted to better understand the associations between individual characteristics and psychological resilience in aging. A developmental approach should reveal changes in the risk profile in old age, the prevalence of some risk factors increases (e.g., deteriorating physical health, cognitive decline), while the impact of other risk factors is attenuated (e.g., family history of anxiety and depression).

The effects of sociodemographic indicators such as sex, age, education, income, and marital status or living situations are still inconsistent across studies. However, some trends can be identified. For example, older women can be more likely to report distress symptoms than men [41–44]. Some researchers have hypothesized that women may be more exposed than men to certain risk factors associated with psychological distress, such as marital stress, parental stress, and domestic stress, and financial stress [45].

Regarding age, studies have found a decrease in psychological distress in older adults compared to younger adults [45–47]. Similar results were found in research that examined the life course of psychological distress: mean levels of distress tended to decrease in adulthood, reaching its lowest point around the 60s, and then increasing without reaching the levels observed in young adults [45]. It should be noted that the oldest segment (70–75 years and older) would be particularly vulnerable to depressive symptoms because they are at greater risk of losing

their functional autonomy, grieving the loss of a life partner, and experiencing a decline in socioeconomic status [22,46,48]. In a review of risk factors for late-life anxiety and depression, Vink et al. [10] found that the odds of negative symptoms in late life appeared to be higher for individuals with lower levels of education, and lower income, as well as those living alone or unmarried (e.g., never married, divorced, or widowed).

Functional autonomy, another important risk factor, is affected by the interaction between normal physiological aging, disease, multimorbidity (i.e., the simultaneous presence of more than one chronic condition), health service utilization, and the costs associated with these services [49]. It may also be influenced by other factors, including physical inactivity, smoking, malnutrition, poverty, social isolation, inappropriate medication use, and falls [50]. Older adults who have lost functional autonomy can experience daily stressors that, among other things, put them at greater risk of depression [51,52].

## Protective factors of psychological distress: Coping and social support

Several researchers have examined the impact of coping with stressful events in aging. Coping frameworks typically include cognitive, behavioral, and emotional dynamics to respond to stressful situations [3,53,54]. According to Lazarus and Folkman [3], most efforts include coping strategies aimed at managing future danger or threat—which is expressed in the efforts one invests in actually changing one's interaction with the environment (problem focus)—and at reducing, preventing, or tolerating the emotional and bodily reactions that are characterized as stressful (emotion focus) [55,56]. Others complete their framework with avoidance strategies [55,57,58].

There is evidence that the use of an avoidance coping style is associated with higher levels of psychological distress [59,60]. Also, older individuals are less likely to experience distress symptoms when they use problem-solving strategies, social support–seeking coping styles, and positive reappraisal [61]. Park et al. [62] examined the moderating effect of coping strategies on the relationship between stress (described as loss experience) and symptoms of depression in a sample of 156 women and 116 men over the age of 60. Men who tended to use a more problem-focused coping style reported fewer depressive symptoms. Among women, those who sought social support in the context of financial stress reported fewer depressive symptoms. However, few studies have been conducted on the coping styles of older people, however. Therefore, little is known about how coping strategies and other relational variables contribute to psychological distress in older adults.

Characteristics of social networks should also be considered when examining potential protective factors for psychological distress in older adults. Numerous researchers have demonstrated the protective effects of both quantitative and qualitative aspects of social networks on mental health in later life [48,63–65]. Quantitative aspects include the structural characteristics of social networks, such as the number of close relationships (network size), the frequency of contact, the type of relationship with the individual (e.g., friendship, close family member), and the availability of the respective individuals for assistance/support in specific contexts (perception of the number of people one can rely on) [66]. The qualitative aspect involves evaluating the support received in terms of satisfaction or dissatisfaction [66,67].

Social relationships have been found to influence mental and physical well-being in later life [68,69]. When a relationship between network size and affect was found (albeit inconsistently), a larger social network was associated with less negative affect, fewer symptoms of depression, and greater well-being [69–72]. Network size is an indicator of more social support resources for individuals. However, some studies have documented that the qualitative aspects of social ties have a stronger influence on mental health or mediate the association between

network size and depressive symptoms or life satisfaction [73–75]. In the presence of detrimental influences, such as stress and a reduced social network for aging individuals, research suggests that higher levels of satisfaction with one's network have a protective effect on mental health [76].

Few studies have been conducted on psychological resilience in aging, and most have focused on a specific type of adversity (e.g., diagnosis or caregiving). A recent review of research on psychological resilience suggested that future studies should aim to consolidate knowledge about the fixed and malleable factors involved. The authors also recommended exploring new indicators that could highlight differential vulnerability to adversity [2]. To our knowledge, no study has examined the joint contribution of sociodemographic factors, coping strategies, and a comprehensive set of indicators related to the interpersonal aspect of psychological distress. In light of stress contagion as an implicit form of adversity, we explore the weights of vulnerability to emotional contagion and empathy in the risk of suffering from psychological distress. The main value of understanding the impact of these additional contributors to the experience of distress is that they are known to be modifiable through psychosocial interventions [77,78].

## Objectives

In the present study, we investigated the level of psychological distress in a sample of older adults in a community setting who were experiencing different types of adversity. We examined whether our sample could be divided into different levels of psychological distress. We then compared the different groups of participants on a range of sociodemographic and psychosocial indicators, including functional autonomy. Finally, while controlling for adversity and pharmacological treatment, we examined whether coping strategies, perceived social support, vulnerability to emotional contagion, and empathy—while also accounting for the more classical factors found in the literature, such as sociodemographic indicators and functional autonomy—could predict the risk of inclusion in clinical groups of psychological distress (see Fig 1 for a summary of the variables interacting within the theoretical model).

In consideration of the associations identified in the existing literature, our hypotheses are as follows:

1. An increase in the use of problem-solving strategies, a reduction in avoidance behaviors, and a greater tendency to seek support may serve to mitigate the risk of psychological distress.

2. The availability of, and higher satisfaction with one's social network may serve to reduce the likelihood of belonging to the subgroup experiencing psychological distress. It is possible that satisfaction, which is a qualitative aspect, may have a more significant impact than availability, which is a quantitative aspect.

3. Those who are particularly susceptible to emotional contagion and empathic processes may be more prone to experiencing psychological distress.

Furthermore, given that the study was primarily conducted in settings accessible to privileged individuals (see the sample characteristics below), the relationships between sociodemographic characteristics and autonomy are exploratory due to the limited information available for this population. Based on several studies, we hypothesize that the aforementioned factors may have a greater impact on explaining psychological distress in this specific sample.

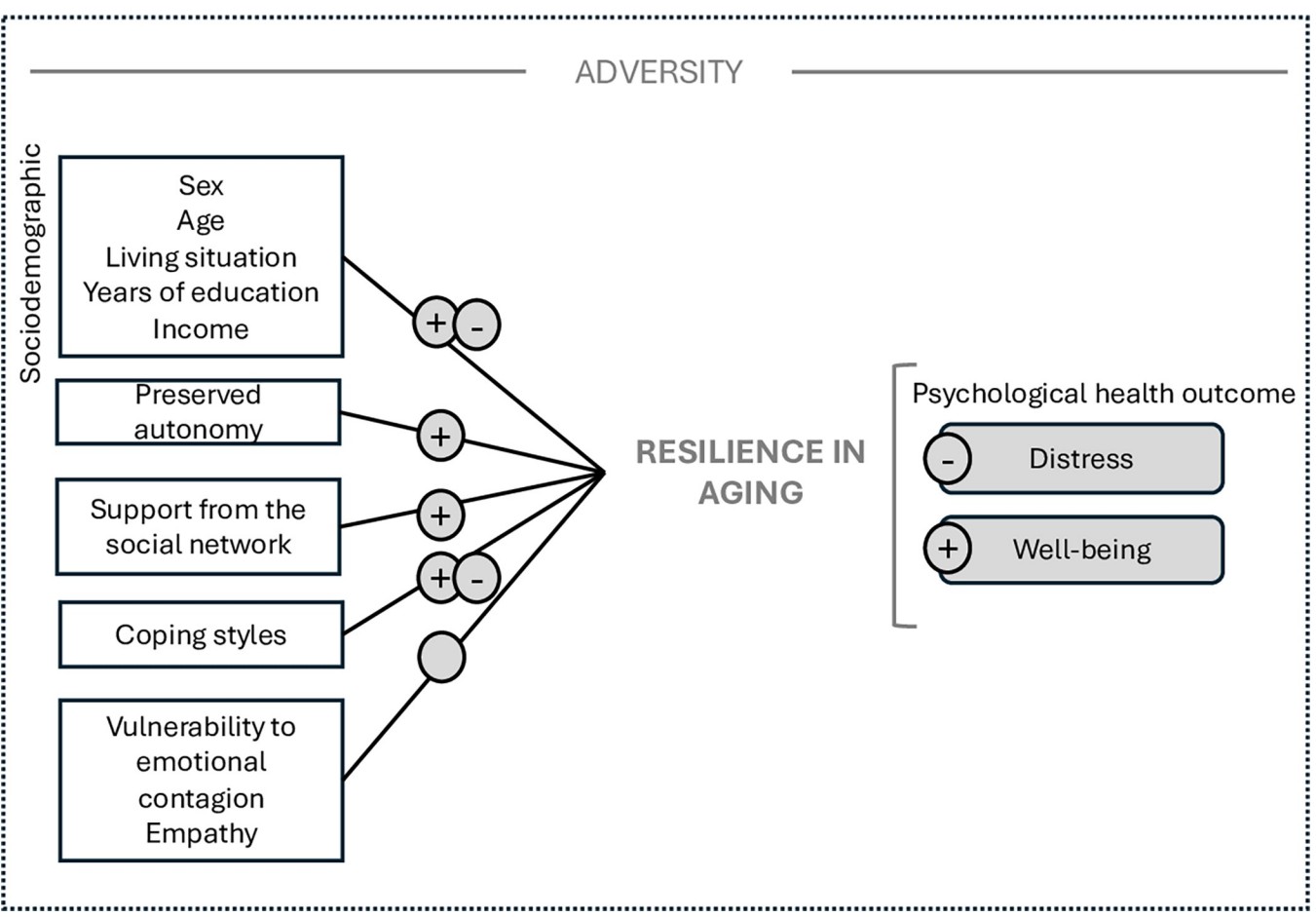

**Fig 1. Overview of the determinants that have been studied in presence of adversity in older age.** The model depicts pathways between the constructs focused–sociodemographic, autonomy, characteristics of a supportive social network, coping styles, vulnerability to emotional contagion, empathy, and psychological health outcome. See text for details.

## Methods

### Participants

This study included participants from a project investigating the effectiveness of a stress management program adapted for older adults (The O'Stress Study, directed by P. Plusquellec) [79,80]. The inclusion criteria for the participants in the original study were: (1) 55 years of age or older; (2) living in a metropolitan area of Quebec, Canada; (3) interested in learning stress management methods in a group setting; and (4) living in a residential home for retirees or frequently participating in activities of organizations serving older adults. Based on a preventive approach, the researchers limited the exclusion criteria to include the general population in terms of levels of distress, life satisfaction, and physical impairment.

### Ethics statement

The study received ethical approval from the University of Montreal Human Research Ethics Committee (No. CERAS-2017-18-018-P). All participants provided written informed consent to participate in the study.

## Measures

A self-report questionnaire was used to collect sociodemographic information and assess perceived functional autonomy. Psychological indicators (distress symptoms, perceived stress, coping strategies, empathy, and emotional contagion) were also measured by self-report questionnaires during group sessions facilitated by the first author of the current study. In addition to the group sessions, one-on-one support (e.g., for individuals with visual impairment or coordination difficulties for tasks like writing) or extra time, and the possibility to complete the questionnaire at home for better concentration with easy access to a PhD student by phone were also offered to the participants according to their needs.

**Psychological distress.**　The Hospital Anxiety and Depression Scale (HADS) was used to measure symptoms of psychological distress. This self-report scale was developed to assess psychological distress, such as symptoms of anxiety and depression, in non-psychiatric patients [81]. The 14 items are scored on a four-alternative response scale ranging from 0 to 3, with 0 indicating the absence of symptoms. Response options differ across questions. Responses are summed after reversing the scores of six questions. The questionnaire is widely used in a variety of settings, including the screening of older adults in the community [82,83]. We used the French version of the questionnaire, which had been validated in a large primary care population, including a population aged 65 years and older [84]. The reliability (Cronbach's alpha) of the global scale in our sample was .83.

**Sociodemographic indicators.**　Information was collected on age, sex, living situation, education level, and income. First, a distinction was made between those living alone and those living with someone (0 = living with someone; 1 = living alone). Participants reported their years of education and household income. Income was divided into six categories ranging from less than 20,000 CAD per year to more than 100,000 CAD per year.

**Functional autonomy.**　Perceived functional limitations were assessed using the short French version of the Assessment of Life Habits (LIFE-H) [85]. The questionnaire measures the performance of eight life habits: communication (eight items), nutrition (four items), housing (eight items), mobility (five items), personal care and health (eight items), social responsibility (eight items), community and spiritual life (eight items), and recreation (seven items). The instrument considers the interaction between personal factors (e.g., disability) and environmental barriers that, when combined, result in functional limitations. For each life habit, the respondent must indicate its level of achievement (4 = without difficulty; 3 = with difficulty; 2 = achieved by substitution; 1 = not achieved; 0 = not applicable) and the "type of help required" to achieve the life habit (4 = no help; 3 = with technical help; 2 = with adjustment; 1 = with someone's help). A normalized score (0–10) is calculated by assigning a weight to each combination of responses. A score of 10 on the global scale indicates the absence of limitations in the realization of one's life habits. The instrument has good psychometric properties in older adults [86–88].

**Coping styles.**　To assess coping styles, we used the Proactive Coping Inventory (PCI) [58]. The PCI measures seven dimensions of coping including problem-focused, emotion-focused, and avoidance strategies. The first five scales are problem-solving types. Proactive Coping includes autonomous goal setting with self-regulatory goal attainment cognitions and behaviors. Reflective Coping describes cognitions that represent simulation and contemplation of comparative action plans to achieve better outcomes. Strategic Planning measures the tendency to generate a goal-directed plan of action in which extensive tasks are broken down into manageable components. Preventive Coping involves strategies that are activated in response to a perceived threat or in a state of worry. Instrumental Support Seeking reflects the reliance on one's social network for help in coping with stressors, including advice and general

information. Emotional Support Seeking is the only scale that measures the emotional aspect of coping. This scale reflects the tendency to self-regulate one's emotions with the help of others by disclosing others' feelings, evoking empathy, and seeking companionship. Avoidance Coping measures the tendency to avoid action in a stressful situation by delaying.

Respondents must rate 55 items on a scale from 1 (not at all true) to 4 (completely true). The scales represent the mean score of the items from 1 to 4. A high score represents a person's strong belief in the potential to change a situation that will result in an improvement to one's situation/environment. The instrument was translated into French by the first author (MJR). In our sample, the psychometric properties were satisfactory, except for the Avoidance Coping scale (three items; alpha = 0.41).

**Social support.**    The short version of the Social Support Questionnaire (SSQ-6) [67] measures both the quantitative and qualitative aspects of social relationships using two scales: Availability and Satisfaction. Availability represents the number of people whom one can rely on when help is needed. Participants are presented with six situations in which they are asked to write down the initials of up to nine people. The sum of the people listed (0–54) forms the Total Availability scale. The Satisfaction scale is described as the perceived adequacy between the support received and the individual's needs and expectations [89]. For the six situations, satisfaction is rated from very dissatisfied (1) to very satisfied (6). The Satisfaction scale is calculated as the mean of the average satisfaction scores. In this study we used the French version, which was translated and validated in five subsamples of adults (average age of 34 years old) [90].

**Vulnerability to emotional contagion.**    The Emotional Contagion Scale (ECS) measures an individual's vulnerability to emotional contagion with five different, basic emotions: love, happiness, fear, anger, and sadness [91]. The score is calculated as the mean of a 15-item questionnaire (e.g., "I get filled with sorrow when people talk about the death of their loved ones"), using a 5-point Likert scale ranging from never (1) to always (5). The original scale was validated with multiple samples and has been used in research to assess the vulnerability of older adults who are caregivers to emotional contagion [92,93]. For our sample, the reliability (Cronbach's alpha) of the French version translated by the research team was .86.

**Empathy.**    We assessed the affective and cognitive aspects of empathy using three scales of the French version of the Interpersonal Reactivity Index [37,94]. Feelings of sympathy and concern for the unfortunate were measured by the Empathic Concern scale. The Personal Distress scale measured feelings of personal anxiety and discomfort in intense interpersonal situations. The cognitive component was assessed with the Perspective Taking scale, which identifies the tendency to spontaneously adopt the psychological point of view of others. Participants rated their agreement or disagreement with each item on a 7-point scale (1 = does not describe me well; 7 = describes me very well), and the mean scores per subscale represented the variables included in the analysis. The scales showed satisfactory internal consistency in our sample (Cronbach's alphas between .59 and .81).

## Control variables

The perception of adversity is a subjective matter. To control for the effect of adversity in our model, we used the Perceived Stress Scale (PSS), a widely used self-report questionnaire [95] validated in French by Ezzati et al. [96]. Mean scores range from 0 to 4, with higher scores indicating greater stress. The reliability (Cronbach's alpha) of the validated French version for our sample was .89. The use of medication for anxiety and depression was also included, as it should directly affect the symptoms of psychological distress. The use of antidepressants and anxiolytics was self-reported.

## Procedure

The present study implemented a cross-sectional analysis using data collected at a baseline period (n = 170). Participants were recruited from private residences and organizations providing services to older adults in a community in Quebec, Canada. Conferences on stress and aging were held between September 10, 2018, and September 8, 2019 to recruit participants. Out of the 264 individuals interested in the study, 170 agreed to participate. The participants ranged in age from 56 to 96 years, with an overall mean age of 76.1 years (SD = 7.7). The sample was predominantly female (85.4%). The majority were born in Canada (87.9%) and lived alone (59.1%). Elementary school was the highest level of education reported by 9.6% of the participants. Nearly as many individuals reported having completed secondary school (44.6%) and post-secondary school education (45.9%). As for income, 15.9% of the participants chose not to provide this information; 11% reported an annual income of $20,000 or less. Most of the sample had annual incomes between $21,000 to $40,000 (30.5%) and $41,000 to $60,000 (22.6%). Finally, 12.1% reported an income above $81,000. In our sample, functional autonomy scores ranged from 5.29 to 10, with a high mean of 9.2 on a scale of 10 (SD = 0.9).

## Missing data

Across the variables, 2.34–9.36% of the data were missing, with a low average mean proportion of 4.7%. After accounting for missing data, 81.17% of the observations were available for analysis. Observations with incomplete data over the three data points were significantly more likely to be in the distress group than in the sample with no missing data ($\chi^2$ = 5.190, p = .023) but were not different for the other variables. In addition, some participants were missing other data on several factors. To handle these missing data, we used the multiple imputation method and calculated 20 datasets based on the distribution of the imputed variables and on the observed data using IBM's SPSS Statistics 29 software procedure. The number of imputed datasets was chosen to be equal to the percentage of incomplete cases [97,98]. All variables from the three measurement points of the original study were included in the imputed model. The setting in which participants were recruited and the experimental condition of the larger study were included in the model as predictors. These 20 datasets were aggregated into one set of parameter estimates.

## Analytic strategy

Analyses were conducted on the imputed data. First, the sample was divided into three groups based on the HADS's clinical cutoff scores for the anxiety and depression subscales. On the mean scales ranging from 0 to 3, participants with scores of 1 and below on both subscales were identified as not experiencing clinical psychological distress. Participants with scores above 1 were identified as individuals reporting subthreshold and clinical anxiety and depression [99]. Differences between the subthreshold and clinical anxiety and depression groups for each independent variable were previously tested by comparing column proportions (chi-squared) or mean differences (t-test). Since the subthreshold and clinical groups did not differ, two profiles were created based on the expression of psychological distress in the participants in our sample. First, individuals with only subthreshold and clinical anxiety symptoms only made up the *anxiety* profile. Second, individuals with subthreshold and clinical depressive and anxiety symptoms made up the *anxious depression* profile. Only three individuals reported subthreshold or clinical levels of depression without anxiety; these were excluded from the analysis.

Next, to address our first objective, we tested whether these three groups differed in terms of sociodemographic characteristics, functional autonomy, and psychosocial factors using chi-squared difference tests (for categorical variables) or one-way analyses of variance (ANOVAs) for continuous variables.

To address our second objective, we used a multinomial logistic regression. Control variables were included in the model, while all other variables were incorporated using a stepwise backward strategy in the analysis. The first multinomial logistic regressions compared the two psychological distress groups with the no-distress group. The second analysis compared the anxiety group with the anxious depression group. Prior to the analysis, multicollinearity was assessed with a linear regression in which the categorical dependent variable was treated as a continuous variable. The Variance Inflation Factor for all predictors was found to be below 10, with values ranging from 1.045 to 1.435. This indicates that the variables are moderately correlated [100]. The significance level was set at $p \leq 0.05$, or confidence interval (CI) = 95%. The data were analyzed using IBM's SPSS Statistics 29 software.

## Results

### Group differences in psychological distress

As assessed by the HADS, 65.9% of our sample reported clinical or subthreshold levels of anxiety and depression (a mean score of greater than 1 for each subscale is indicative of an overall score of 8 and above on a scale of summed scores from 0 to 21). Table 1 presents the descriptive statistics for each study variable for the global sample and for the no-distress, with anxiety, and with anxious depressive symptoms groups. With respect to sociodemographic characteristics, no group differences were found for age ($p$ = .356), education ($p$ = .693), living situation ($p$ = .358), and income ($p$ = .841). A chi-squared test of independence revealed a marginal association between sex and the psychological distress groups, $X^2(2)$ = 5.412, $p$ = .067. The observed trend concerned a slightly higher than expected proportion of males in the anxious depression group. Regarding the perception of adversity, all groups differed in perceived stress ($p$ < .000). Individuals with no distress symptoms reported significantly less stress than the other two groups. Individuals with only anxiety symptoms also reported less stress than those in the anxious depression group, $p$ < .001, 95% CI = [-.766, -.283]. Finally, the results showed that the use of medication to treat anxiety or depression was higher than expected for individuals in the anxious depression group ($p$ < .001).

One-way ANOVAs were performed to compare the effects of all independent variables on the different distress groups (Table 1). A statistically significant difference in functional autonomy was found between at least two groups (see Table 1). Post hoc pairwise comparisons using Tukey's HSD indicated that the no-distress group was marginally different from the anxiety group ($p$ = .059) and significantly different from the anxious depression group ($p$ < .001), which reported lower autonomy than the anxious group ($p$ = .044). In terms of emotional contagion, the anxiety only ($p$ < .0001 and the anxious depression ($p$ = .001) groups showed greater vulnerability to emotional contagion compared to the no-distress group. On the four empathy scales, the only group difference was on the empathic distress subscale, where participants in the no-distress group reported significantly less empathic distress than the other two groups ($p$ < .001). In addition, participants in the no-distress group reported significantly higher mean satisfaction with their social network than those in the anxious ($p$ = .008) and anxious depressed ($p$ = .001) groups.

An analysis of coping strategies revealed differences between groups, except for avoidance and seeking instrumental support. Individuals with only anxiety symptoms did not differ from those without distress in coping strategies. However, those with anxious depression showed significant differences from the no-distress group in proactive ($p$ = .001), reflective ($p$ = .048), strategic planning ($p$ = .041), preventive ($p$ = .022), and emotional support ($p$ = .014) strategies but not in preventative coping strategies ($p$ = .459). Differences were also found between the anxious depression and anxiety groups in proactive ($p$ = .004), reflective ($p$ = .011), strategic

**Table 1. Descriptive statistics for the global sample and for the no-distress, anxiety, and anxious depression subgroups.**

| | Global sample | No distress | Anxiety | Anxious depression | Test statistic |
|---|---|---|---|---|---|
| | Mean (SD) / % | Mean (SD) / % | Mean (SD) / % | Mean (SD) / % | |
| | N = 170 | N = 57 | N = 75 | N = 35 | |
| Sex (being male) | 15.0% | 19.3% | 8.0% | 22.9% | $\chi^2(2, 167) = 5.412^t$ |
| Age | 76.12 (7.72) | 75.73 (7.75) | 77.01 (7.45) | 74.75 (8.22) | $F(2, 164) = 1.038$ |
| Cohabitation | 40.7% | 36.8% | 46.7% | 34.3% | $\chi^2(2, 167) = 2.054$ |
| Income | | | | | $\chi^2(10, 167) = 5.680$ |
| < $20,000 | 12.6% | 8.8% | 12.0% | 20.0% | |
| $21,000–40,000 | 32.9% | 29.8% | 33.3% | 37.1% | |
| $41,000–60,000 | 28.1% | 31.6% | 29.3% | 20.0% | |
| $61,000–80,000 | 12.0% | 15.8% | 10.7% | 8.6% | |
| $81,000–100,000 | 6.6% | 7.0% | 5.3% | 8.6% | |
| > $101,000 | 7.8% | 7.0% | 9.3% | 5.7% | |
| Years of education | 13.64 (3.51) | 13.89 (3.42) | 13.38 (3.62) | 13.78 (3.47) | $F(2, 164) = .369$ |
| Control | | | | | |
| Psychotropic agents (use = 1) | 39.5% | 22.8% | 40.0% | 65.7% | $\chi^2(2, 167) = 16.716^{***}$ |
| Perceived stress | 1.65 (.62) | 1.20 (.47) | 1.71 (.56) | 2.23 (.41) | $F(2, 164) = 47.028^{***}$ |
| Functional autonomy | 9.15 (.88) | 9.46 (.50) | 9.12 (.95) | 8.70 (1.05) | $F(2, 164) = 8.845^{***}$ |
| Emotional Contagion Scale | 3.51 (.56) | 3.19 (.48) | 3.72 (.52) | 3.61 (.55) | $F(2, 164) = 17.662^{***}$ |
| Empathy | | | | | |
| Perspective taking | 4.76 (.80) | 4.79 (.83) | 4.78 (.82) | 4.69 (.75) | $F(2, 164) = .178$ |
| Empathic concern | 5.55 (.83) | 5.42 (.79) | 5.64 (.92) | 5.54 (.68) | $F(2, 164) = 1.178$ |
| Empathic distress | 3.65 (1.19) | 2.93 (1.13) | 3.98 (1.10) | 4.11 (.92) | $F(2, 164) = 19.166^{***}$ |
| Social network | | | | | |
| Availability | 19.22 (9.94) | 21.55 (10.42) | 18.21 (8.10) | 17.60 (12.11) | $F(2, 167) = 2.467^t$ |
| Satisfaction | 4.94 (1.23) | 5.41 (.86) | 4.77 (1.41) | 4.49 (1.15) | $F(2, 167) = 7.633^{***}$ |
| Coping style | | | | | |
| Proactive | 2.75 (.31) | 2.81 (.27) | 2.78 (.31) | 2.58 (.36) | $F(2, 167) = 7.307^{***}$ |
| Strategic planning | 2.74 (.51) | 2.79 (.46) | 2.81 (.52) | 2.53 (.55) | $F(2, 167) = 4.174^{**}$ |
| Reflective | 2.86 (.35) | 2.88 (.32) | 2.91 (.29) | 2.71 (.47) | $F(2, 167) = 4.470^{**}$ |
| Preventive | 2.97 (.38) | 2.94 (.37) | 3.05 (.36) | 2.85 (.43) | $F(2, 167) = 3.867^{*}$ |
| Instrumental support seeking | 2.81 (.46) | 2.76 (.45) | 2.86 (.43) | 2.75 (.53) | $F(2, 167) = 1.063$ |
| Emotional support seeking | 2.89 (.51) | 2.94 (.47) | 2.97 (.46) | 2.64 (.61) | $F(2, 167) = 5.772^{**}$ |
| Avoidance | 2.81 (.49) | 2.82 (.49) | 2.77 (.53) | 2.86 (.40) | $F(2, 167) = .378$ |

Note.

$^*p < .05$

$^{**}p < .01$

$^{***}p < .001$.

planning ($p = .018$), preventive ($p = .022$), and emotional support ($p = .004$) strategies. Overall, individuals with anxious depression symptoms used proactive, reflective, strategic planning, preventive, and emotional support strategies less than the other two groups.

## Variables contributing to psychological distress profiles

A multinomial logistic regression analysis was performed with the three psychological distress groups as the dependent variable. Control variables, sociodemographic characteristics (sex, age, income, years of education, living situation), functional autonomy, coping styles, social

network characteristics, vulnerability to emotional contagion, and empathy were included as independent variables in the model.

The model fit of the final model was assessed using the goodness-of-fit test, and it indicated a good fit to the data, $\chi^2(14; N = 167) = 139.146, p < .001$. The chi-square test also indicated that the final model explained a significant amount of the original variability. When controlling for adversity and the use of psychotropic treatment, the results revealed that sex, satisfaction toward network support, emotional support–seeking and avoidance coping styles, and vulnerability to emotional contagion significantly predicted the likelihood of presenting no distress, anxiety, or anxious depression symptoms. The overall model accounted for a significant amount of variance in the psychological distress groups (Nagelkerke's $R^2 = .643$). Table 2 shows the coefficients, the Wald statistics, associated degrees of freedom, and probability values for each of the variables included in the final model. Specifically, the first analysis used the no-distress group as the reference group, and the second analysis used the anxious depression group.

Perceived stress as a measure of adversity and the use of a psychotropic treatment showed significant associations with the outcome variable. Perceived stress was identified as a significant predictor of anxiety ($p < .001$) and anxious depression ($p < .001$), while psychotropic treatment was identified as a significant predictor of belonging to the anxious depression group only ($p = .019$). An odds ratio calculation indicated that a 1-point increase on the perceived stress scale was associated with an increase in the odds of being in the anxiety only group by a factor of 6.330 and in the anxious depression group by a factor of 88.793.

After controlling for perceived stress and the use of psychotropic treatment, only the variable of sex remained as a significant predictor in the final model; all other sociodemographic variables were no longer significant. The probability of belonging to the anxious depression subgroup was found to be greater for males ($p = .014$) compared to the group without symptoms. Furthermore, the relationship between vulnerability to emotional contagion and psychological distress remained statistically significant for anxiety ($p < .001$) and anxious depression ($p < .001$) when compared to individuals with no reported distress symptoms. For each unit increase on the vulnerability to emotional contagion scale, the odds of presenting anxiety and anxious depression increased by factors of 12.346 and 12.574, respectively. Moreover, social network satisfaction was identified as a significant predictor of belonging to the anxiety and anxious depression subgroups ($p = .010$ and $p = .032$, respectively). Individuals who were less satisfied with their social network were more likely to be in the anxiety and the anxious depression subgroups than in the no-distress group. With regard to coping styles, individuals who reported a reduced tendency to utilize avoidance coping strategies were more likely to be in the anxious ($p = .006$) and anxious depression group ($p = .027$) than be individuals without distress.

In comparing the two distress profiles, we found that seeking emotional support appeared to play a significant role ($p = .010$). Individuals with anxiety were more likely to turn to others to help them regulate their emotions compared to individuals with anxious depression. Finally, men were more likely to fit the anxious depression symptoms profile than the anxiety symptoms profile ($p = .024$).

## Discussion

Although adversity is a normal part of life, the ability to cope with ongoing—and sometimes persistent—stressors is essential for maintaining mental health. Aging can exacerbate the challenges of the transitions and changes experienced in one's life and body. The severity of the consequences of psychological distress calls for a better understanding of the factors involved, facilitating the identification of those at risk and guiding practice to prevent these consequences.

**Table 2. Multinomial stepwise logistic regression for psychological distress groups.**

| Anxiety vs. No distress | | | | | | |
|---|---|---|---|---|---|---|
| | B | SE | Wald | Exp(B) | 95% CI | |
| Control | | | | | | |
| Perceived stress | 1.845 | .517 | 12.761 | 6.330 | [2.300, | 17.423]*** |
| Psychotropic agents | .739 | .531 | 1.934 | 2.093 | [.739, | 5.928] |
| Sociodemographic | | | | | | |
| Sex | .492 | .702 | .492 | 1.636 | [.413, | 6.477] |
| Emotional Contagion Scale | 2.513 | .595 | 17.860 | 12.346 | [3.849, | 39.605]*** |
| Social network | | | | | | |
| Satisfaction | -.675 | .261 | 6.709 | .509 | [.305, | .848]** |
| Coping styles | | | | | | |
| Emotional support seeking | .509 | .538 | .892 | 1.663 | [579, | 4.777] |
| Avoidance | -1.386 | .506 | 7.506 | .250 | [.093, | .674]** |
| **Anxious depression vs. No distress** | | | | | | |
| | B | SE | Wald | Exp(B) | 95% CI | |
| Control | | | | | | |
| Perceived stress | 4.486 | .795 | 31.843 | 88.793 | [18.692, | 421.802]*** |
| Psychotropic agents | 1.665 | .708 | 5.530 | 5.285 | [1.320, | 21.164]* |
| Sociodemographic | | | | | | |
| Sex | 2.335 | .950 | 6.039 | 10.325 | [1.604, | 66.449] |
| Emotional Contagion Scale | 2.532 | .738 | 11.772 | 12.574 | [2.961, | 53.398]*** |
| Social network | .246 | .340 | .524 | 1.279 | [.657, | 2.491] |
| Satisfaction | | | | | | |
| Coping strategies | | | | | | |
| Emotional support seeking | -.983 | .740 | 1.765 | .374 | [.088, | .1.596] |
| Avoidance | -1.532 | .694 | 4.882 | .216 | [.055, | .841]* |
| **Anxiety vs. Anxious depression** | | | | | | |
| | B | SE | Wald | Exp(B) | 95% CI | |
| Control | | | | | | |
| Perceived stress | -2.641 | .644 | 16.815 | .071 | [.020, | .252]*** |
| Psychotropic agents | -.926 | .550 | 2.836 | .396 | [.135, | 1.164] |
| Sociodemographic | | | | | | |
| Sex | -1.842 | .818 | 5.073 | .158 | [.032, | .787]* |
| Emotional Contagion Scale | -.018 | .504 | .001 | .982 | [.366, | 2.635] |
| Social network | | | | | | |
| Satisfaction | .028 | .232 | .015 | 1.028 | [.653, | 1.619] |
| Coping strategies | | | | | | |
| Emotional support seeking | 1.492 | .577 | 6.688 | 4.445 | [1.435, | 13.768]** |
| Avoidance | .147 | .547 | .072 | 1.158 | [.396, | 3.385] |

Note.

*$p < .05$

**$p < .01$

***$p < .001$.

Reviews have suggested exploring new factors associated with the process of psychological resilience. Given the different nature of explicit and implicit adversity, it seemed appropriate to test the impact of vulnerability on implicit stressors. Therefore, this study aimed to test the differential weights of a wide range of factors, including sociodemographic factors, indicators

of autonomy, social support, coping styles, vulnerability to emotional contagion, and empathy in the presence or absence of two profiles of psychological distress. Four main findings merit discussion.

First, consistent with previous studies in community settings, our sample included a high proportion of individuals with subthreshold and clinical anxiety and depression according to scores on the Hospital Anxiety and Depression Scale. This portrait is congruent with the aim of the larger study, which was to recruit individuals to test the effect of an intervention on stress management and stress-related mood disorders. Some of the recruitment environments in this study also included people who were more vulnerable due to some specific risks, such as caregiving, social isolation, and chronic illness. In the context of this study, participants may have felt more comfortable disclosing distress symptoms. Both profiles—anxiety only and anxious depression—included a larger proportion of people with subthreshold symptoms than with clinical symptoms. Subthreshold symptoms are known to be common in older adults and are associated with similar adverse outcomes as clinical levels of anxiety and depression [11,101,102]. Our sample included very few individuals with only symptoms of depression, which is consistent with previous research that found a low prevalence of depression per se in the aging population [41]. It is also possible that the group intervention proposed in the main study may have discouraged people with a more depressive profile from participating (e.g., social withdrawal, low self-care); moreover, somatic symptoms such as malaise and even cognitive complaints with memory loss and attention deficit may discourage an older adult from participating in the study. The stressful context of a person with severe distress symptoms (e.g., caring for a partner who presents severe behavioral symptoms of Alzheimer's disease) may affect their ability to seek help. Studies have reported that more than half of the cases of depression in older adults evidence anxiety symptoms [9,103]. Thus, the co-occurrence of symptoms from both conditions has been associated with a more chronic course of distress [41]. The different patterns of risk factor profiles and disease trajectories found in the literature support the subsequent comparison of these expressions of psychological distress.

Second, as expected, individuals with both anxiety and anxious depression profiles reported more adversity, as measured by the Perceived Stress Scale, than those without distress. Moreover, individuals with anxious depression also perceived higher levels of stress in their current lives compared to individuals with an anxiety profile. Other studies have found direct associations between perceived stress and depression and anxiety in older adults [104].

No significant results were found when comparing the psychological distress groups and the no-distress group based on sociodemographic variables such as age, education, income, and living situation. Again, the association between sociodemographic indicators and psychological distress showed some tendencies, although they are not replicated in other studies. The work of Cole and Dendukuri [105], Djernes [106], and Vink et al. [10] illustrates the variability found in studies of the associations between psychological distress and a variety of sociodemographic factors in older adults. A study of 236,503 older adults in Australia found that the overall prevalence of psychological distress was lower at older ages, whereas there was a gradual increase in the proportion of high or very high scores from both men and women aged 80 years or older, [22]. In a German sample of 1,659 individuals aged from 60 to 85 years, researchers found that depressive symptoms were significantly associated with increasing age in both men and women [48].

In our study, the lack of association may be explained by a recruitment bias. In fact, more than 75% of our sample lived in high-end private residences. In addition to gaining comfort and security, living in a residence does include its own distinct stressors, such as recent relocation and the cost of living [107]. Individuals living in a retirement home, especially a private one, represent a rarely studied population because they are not as concerned about typical risk

factors (i.e., they are individuals with higher incomes and education levels, with a safety net of services provided by the living situation). As previously stated, the observed variability in the sociodemographic variables included in this study is not comparable to that seen in other studies due to the distinctive characteristics of the recruitment sites. Our results indicate that being male was associated with an increased likelihood of being in the anxious depression group compared to the anxious group. The prevalence of depression and anxiety has been shown to vary between studies, with sex differences being a prominent factor. However, the limited representation of men in the present study reduces the generalizability of the conclusions that can be drawn. However, this finding raises the question of whether the men in our population were more at risk of presenting distress with a combination of anxious and depressive symptoms. Given the context of our study, one might also speculate that the men may wait until they experience more severe symptoms, as expressed with an anxious depressive profile, before they seek professional help.

Third, vulnerability to emotional contagion emerged as the primary factor that impacted the likelihood of being in both psychological distress groups, relative to individuals with no distress. After controlling for adversity and psychotropic treatment, vulnerability to emotional contagion had the strongest relationship with both psychological distress profiles. Individuals who reported being more susceptible to unconsciously catching the emotions of others were more likely to be in the anxiety or anxious depression group. In light of the current understanding of HPA axis activation in response to explicit and implicit stressors, our hypothesis is that individuals who are more sensitive to their emotional environment are likely to experience greater activation of their HPA axis. This heightened response may be challenging to manage and generate a toll on a vulnerable stress system with age. Increased exposure to stress hormones due to both explicit and implicit stress, coupled with an altered HPA axis, could elevate the risk of distress symptoms in this population—particularly if individuals struggle to identify the source of their stress, which leads to a greater sense of lack of control. Conversely, individuals experiencing distress symptoms could be more vulnerable to the negative emotions in their environment [108,109].

In older populations, emotional contagion is typically studied in the context of caregiving or marital relationships. These contexts usually include the examination of affect contagion (i.e., how the presence of depressive affect in one partner influences the risk of depressive symptoms in the other partner). In the present study, we chose to examine emotional contagion in terms of vulnerability. To our knowledge, there were no data on vulnerability to emotional contagion in older adult populations. Yet, vulnerability to emotional contagion has been associated with mental health outcomes in other populations. For example, one study found that vulnerability to emotional contagion among physicians and nurses influenced job burnout, depending on the source of the contagion (i.e., peers, leaders, or patients) [110]. Some research on caregiving and emotional contagion has also highlighted the mental health risks of these close interactions [111,112]. The scores on the emotional contagion susceptibility scale in our sample (M = 3.51; SD = 0.567) coincided with findings from other studies of younger populations [91,113]. Our findings suggest that among community-dwelling older adults, the vulnerability to emotional contagion trait can be associated with a higher risk for one's mental health. It is worth mentioning that even though empathic distress, as assessed by the Interpersonal Reactivity Index, was significantly lower in the no-distress group compared to both distress groups, it did not affect the likelihood of belonging to either group when all variables were considered. This means that vulnerability to the unconscious aspects of sharing one's emotions had a stronger relationship with mental state, even when the conscious aspect of affective and cognitive empathy was considered, although the study design did not allow us to pinpoint the dynamic between a higher vulnerability to emotional contagion and psychological distress.

However, our findings highlight the role of the social environment in understanding and improving mental health. In several populations, studies have shown that mood (negative and positive) can easily be "caught" from others in a variety of indirect and direct interactions such as looking at pictures, watching a short video clip, and scrolling through social media content [114,115]. Caregiving relationships [112] and providing brief emotional support to a troubled friend [116] are examples of specific contexts conducive to emotional contagion, especially for women. To a greater extent, as detailed by Horesh et al. [117], contagion of psychiatric disorders such as depression has also been observed. These studies and our findings confirm the importance of a contextual and environmental perspective, alongside a person-centered approach, for understanding psychological distress. In the specific context of our study, where the participants mostly lived in retirement homes, it seems essential to examine the dynamics in the interpersonal environments of these settings and how they can influence mental health both positively and negatively. Also, one should question the similarity between the possible consequences of adversity in chronic stress and an implicit stressor such as stress contagion when an individual is vulnerable and highly exposed to negative affect.

In our model, participants who reported more satisfaction with their social network were less likely to report anxiety or anxious depression. Consistent with findings in other studies, the qualitative aspect of the social network played a more important role in mental health than the quantitative aspect. For older adults, satisfaction with one's network is more often a resource factor than having to do with the size of the network available [66,118,119]. In the presence of adversity, the network is known to play a central role by (1) modulating the interpretation of the challenge faced so that the appraisals are less negative, (2) suppressing the cortisol stress-related response, and (3) influencing appraisals of self and others in the face of adversity, thus stimulating the use of coping strategies more adapted to the situation [120–122]. Less than a month after the completion of our study, the COVID-19 pandemic occurred, highlighting the detrimental effects of isolation on all populations—but especially on older adults [123]. Many social support programs have proven effective in reducing psychological distress in older adult populations [124,125]. Some authors have also recommended taking a public health approach to social isolation to improve their physical and mental health of older adults [126,127].

Fourth, individuals with anxiety and those with anxious depression differed on two coping factors: the tendency to seek emotional support and the use of avoidance. Compared to individuals with anxious depression, individuals with anxiety tend to seek emotional support from their network more often when faced with adversity. This finding could be explained by several factors such as a mismatch between the support available and the needs of the individuals or recall bias, as explained above. The higher proportion of men in the anxious depression group may have influenced the effect of these coping strategies as well. Some studies comparing the use of coping strategies between men and women have reported that women are more likely than men to seek emotional support [62,128]. Undoubtedly, these observations would have been better supported by the presence of other indicators based on previous findings, such as history of mental health, alcohol use, and the presence of a life-threatening illness [129]. Finally, our models did include the avoidance coping strategy. The direction of the relationship is contrary to that observed in the literature, with a greater likelihood of being in the clinical groups when using fewer avoidance strategies. The majority of studies have found a correlation between avoidance coping and elevated physiological stress and distress levels [59,60,130]. The poor psychometric quality of the measure precludes any meaningful interpretation. The avoidance scale from the Proactive Coping Inventory, designed to assess the tendency to defer action in stressful situations, has demonstrated cultural nuances in its translation. Two items

suggest a "composed restraint", while one indicates "evasion," which aligns with the authors' definition. It is therefore proposed that individuals who pause before responding to stressors may exhibit a lower likelihood of being in the clinical groups, potentially indicating a form of emotional regulation. However, it should be noted that our research did not delve into specific emotional regulation strategies.

Our findings lead to new implications for older adults. First, it may be appropriate to examine the key moderators of individual trait vulnerability to emotional contagion in older populations (i.e., sex, age, attention to others, perception of self as interdependent with others, frequent imitation of others' facial, vocal, and postural expressions, and conscious emotional experience that is strongly influenced by social cues and feedback), supported by the work of Hatfield et al. in their theory of emotional contagion and by others who have contributed to our current understanding of emotional contagion [29,131–133]. Second, our findings suggest that it may be useful to examine the effectiveness of psychological interventions aimed at teaching strategies for managing emotional contagion to prevent psychological distress in older adults. Researchers have developed programs or suggested strategies to better cope with stress or reduce the negative effects of emotional contagion [78,134]. It would be interesting to evaluate whether these strategies, adapted to an older population, can prevent psychological distress.

The cross-sectional design of this study did not allow for speculation on causation. Although the original study used repeated measures over 3 months, the program evaluation did not recruit enough eligible subjects for this secondary analysis. A prospective longitudinal study may provide an opportunity to explain the direction of the relationship between psychological distress and emotional contagion, empathy, coping, and social support. In addition, selection bias in the context of evaluating an intervention might limit the generalizability of the results. A major strength of this study, however, was the inclusion of sociodemographic and psychosocial variables in a model, including original indicators such as emotional contagion and empathy. It should be noted that the inclusion of additional variables, such as comorbidity and a history of mental health problems, could have provided further insight into the observed phenomenon. To our knowledge, our study was the first to examine the relationship between psychological distress and vulnerability to emotional contagion in older adults. Thus, replications of this study might prove useful.

## Conclusion

The consequences of psychological distress in an older population can be harmful. To date, studies have identified many factors to better understand the mechanisms and individual characteristics that increase the risk of psychological distress. The present study showed that the older adults who were less satisfied with the social support they received, and those who reported higher vulnerability to emotional contagion were more likely to report subthreshold and clinical symptoms of anxiety and depression. Also, the findings described in this study touched on factors that are rarely studied with this population, contributing to a broader understanding of the impact of individual ties with social connections. Finally, our results support the value of interventions like programs aimed at improving satisfaction with one's social network and enhancing the cognitive mastery of emotional contagion to reduce or prevent psychological distress in the growing aging populations.

## Supporting information

**S1 Appendix. Additional analysis on the depression and anxiety factors of Hospital Anxiety and Depression Scale (HADS) as a continuous outcome.**
(DOCX)

## Acknowledgments

We would like to express our heartfelt gratitude to Le Groupe Maurice, the Association des Personnes Aidantes de Laval, the Fondation Berthiaume-du-Tremblay, especially the Centre de Jour, and the participants of the study who provided important information for this research. We thank the Groupe Maurice whose grant supported this study.

## Author Contributions

**Conceptualization:** Marie-Josée Richer, Sébastien Grenier, Pierrich Plusquellec.

**Data curation:** Marie-Josée Richer.

**Formal analysis:** Marie-Josée Richer.

**Funding acquisition:** Marie-Josée Richer, Sébastien Grenier, Pierrich Plusquellec.

**Methodology:** Marie-Josée Richer, Pierrich Plusquellec.

**Project administration:** Marie-Josée Richer.

**Supervision:** Pierrich Plusquellec.

**Validation:** Pierrich Plusquellec.

**Writing – original draft:** Marie-Josée Richer.

**Writing – review & editing:** Sébastien Grenier, Pierrich Plusquellec.

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
