## [Decision Letter · Decision Letter 0]

10 Jun 2024

PMEN-D-24-00145

The contribution of vulnerability to emotional contagion to the expression of psychological distress in older adults

PLOS Mental Health

Dear Dr. Richer,

Thank you for submitting your manuscript to PLOS Mental Health. After careful consideration, we feel that it has merit but does not fully meet PLOS Mental Health’s publication criteria as it currently stands. Therefore, we invite you to submit a revised version of the manuscript that addresses the points raised during the review process.

We look forward to receiving your revised manuscript.

Kind regards,

Bochra Nourhene Saguem, M.D.

Academic Editor

PLOS Mental Health

Journal Requirements:

1. In the online submission form, you indicated that "Given the small number of participants and the risk of re-identification associated with small samples, the data set will not be made public. However, the data will be made available on request to the project's principal investigator". 

3. Uploaded as supplementary information.

Additional Editor Comments (if provided):

Reviewers' comments:

Reviewer's Responses to Questions

**Comments to the Author**

1. Does this manuscript meet PLOS Mental Health’s publication criteria? Is the manuscript technically sound, and do the data support the conclusions? The manuscript must describe methodologically and ethically rigorous research with conclusions that are appropriately drawn based on the data presented.

Reviewer #1: Yes

Reviewer #2: Yes

2. Has the statistical analysis been performed appropriately and rigorously?

Reviewer #1: No

Reviewer #2: Yes

3. Have the authors made all data underlying the findings in their manuscript fully available (please refer to the Data Availability Statement at the start of the manuscript PDF file)?

Reviewer #1: Yes

Reviewer #2: Yes

4. Is the manuscript presented in an intelligible fashion and written in standard English?

Reviewer #1: Yes

Reviewer #2: Yes

5. Review Comments to the Author

Reviewer #1: The study examines a cross-sectional sample of older Canadian adults by profiles of psychological distress, and examines the different factors and indicators that differentiate these profiles. Particularly, they found that vulnerability to emotion contagion and social network satisfaction emerged as two interesting predictors of distress in their sample. Overall the study is conducted well and well written. I enjoyed reading the manuscript, although I have some concerns about the analytic methods employed, and some suggestions to improve the flow and readability of the manuscript.

Major comments:

1. The introduction of the different components of the model was largely well written and provided a good overview of the authors’ reasons of examining these components in older adults. One suggestion for improvement would be better signposting, such as the use of sub-headers to clearly state the different theoretical components of the model. Additionally, I felt the section on vulnerability to emotional contagion a bit short, and I wasn’t able to figure out a clear definition from the authors. As it was it felt somewhere in-between stress contagion and empathy, but further clarification from the authors on it’s distinct theoretical relevance would be very helpful.

2. For the comparisons across the different components and indicators, on one hand these appeared to be theoretical, as the authors seem to have some inkling of direction when introducing them. Yet, the results and discussion appear to be written in a speculative and exploratory manner. This is not wrong by any sense, but it would to have some consistency between the introduction and discussion on this – if this was theoretically driven, could the authors specify some expected (hypothesized) relationships in the introduction/objective section, or if this was exploratory, could the authors explicitly state the exploratory nature of the research early on?

3. I appreciate the authors’ intention to use cutoffs for anxiety and depression from HADS for segmentation into profiles. I’m also curious to know if at a construct level, if these would also show the same trends as identified through the multinomial logistic regressions – if looking purely at the depression and anxiety factors of HADS, would OLS regressions conducted on the similar models with these HADS factors as continuous outcome variables reveal similar associations with say functional autonomy? I think these would be useful to have in a supplementary material as an additional analysis, as I imagine certain other disciplines of study would be interested in these kinds of results, as opposed to using cutoffs.

4. It seems that the authors used a multinomial logistic regression with all variables, before excluding predictor variables with p-values above 0.20 as a way to simplify the model and reduce or exclude extraneous predictor variables. I don’t think this is the most appropriate measure to use, given that significance and variance may change depending on the inclusion and exclusion of variables (multicollinearity). I’d recommend the authors look at penalty-based (regularization) regression models like ridge or lasso regressions, or (although still somewhat subjective) hierarchical or stepwise regressions instead that look at significant changes between addition or subtraction of variables or changes in R2.

5. In my opinion, several of the results presented are very much expected (such as the relationship between anxiety/depression and adversity), but the finding on vulnerability to emotion contagion and distress was quite interesting. I wonder if this could be written up in such a way in the discussion that gave it slightly more prominence as one of the key findings in this paper. This would also mean that the introduction could somewhat ‘foreshadow’ this finding by elaborating on possible mechanisms between vulnerability to emotion contagion (see above comment) and distress. Again, this is just a small suggestion, and frankly the discussion reads fine as it currently is to me – although I didn’t quite get the link between proprioceptive sensitivity, and vulnerability to emotion contagion and distress.

Minor comments:

1. I appreciate the objective section of the introduction summarizing the focus of the study and the way the different variables interact within the theoretical model. Could this also be visualized in some way to make it even easier for the reader to understand?

Please give a citation or a reference to the O’Stress study if possible.

2. It seems that participants in the distress group were more likely to have incomplete data. Could the authors comment on this in the discussion section?

3. Could p < .000 be changed to p < .001.

Reviewer #2: This cross-sectional study provided novel insights into vulnerability factors like emotional contagion in psychological distress among older adults. However, some issues need to be discussed:

The introduction section is too long and needs to be summarized. Additionally, labelling in the introduction should be removed.

There were missing details on some variables like comorbid medical conditions that could impact psychological distress.

The mechanisms linking vulnerability to emotional contagion and psychological distress were not directly examined.

6. PLOS authors have the option to publish the peer review history of their article (what does this mean?). If published, this will include your full peer review and any attached files.

**Do you want your identity to be public for this peer review?** For information about this choice, including consent withdrawal, please see our Privacy Policy.

Reviewer #1: No

Reviewer #2: No

---

## [Decision Letter · Decision Letter 1]

23 Sep 2024

The contribution of vulnerability to emotional contagion to the expression of psychological distress in older adults

PMEN-D-24-00145R1

Dear Dr. Richer,

We are pleased to inform you that your manuscript 'The contribution of vulnerability to emotional contagion to the expression of psychological distress in older adults' has been provisionally accepted for publication in PLOS Mental Health.

Best regards,

Bochra Nourhene Saguem, M.D.

Academic Editor

PLOS Mental Health

Reviewer Comments (if any, and for reference):

Reviewer's Responses to Questions

**Comments to the Author**

1. If the authors have adequately addressed your comments raised in a previous round of review and you feel that this manuscript is now acceptable for publication, you may indicate that here to bypass the “Comments to the Author” section, enter your conflict of interest statement in the “Confidential to Editor” section, and submit your "Accept" recommendation.

Reviewer #1: All comments have been addressed

Reviewer #2: All comments have been addressed

2. Does this manuscript meet PLOS Mental Health’s publication criteria? Is the manuscript technically sound, and do the data support the conclusions? The manuscript must describe methodologically and ethically rigorous research with conclusions that are appropriately drawn based on the data presented.

Reviewer #1: Yes

Reviewer #2: Yes

3. Has the statistical analysis been performed appropriately and rigorously?

Reviewer #1: Yes

Reviewer #2: Yes

4. Have the authors made all data underlying the findings in their manuscript fully available (please refer to the Data Availability Statement at the start of the manuscript PDF file)?

Reviewer #1: Yes

Reviewer #2: Yes

5. Is the manuscript presented in an intelligible fashion and written in standard English?

Reviewer #1: Yes

Reviewer #2: Yes

6. Review Comments to the Author

Reviewer #1: (No Response)

Reviewer #2: (No Response)

7. PLOS authors have the option to publish the peer review history of their article (what does this mean?). If published, this will include your full peer review and any attached files.

**Do you want your identity to be public for this peer review?** For information about this choice, including consent withdrawal, please see our Privacy Policy.

Reviewer #1: No

Reviewer #2: No
